# A Data-Driven Approach for Rathke’s Cleft Cysts Management

**DOI:** 10.3390/diagnostics15070886

**Published:** 2025-04-01

**Authors:** Alberto D’Amico, Martina Cappelletti, Nicola Bresolin, Elisabetta Marton, Luca Denaro, Giuseppe Canova

**Affiliations:** 1Academic Neurosurgery, Department of Neurosciences, University of Padova, 35128 Padova, Italy; nicola.bresolin@studenti.unipd.it (N.B.); luca.denaro@unipd.it (L.D.); 2Neurosurgery Unit, AULSS2 Marca Trevigiana, Treviso Hospital, 31100 Treviso, Italy; martina.cappelletti@aulss2.veneto.it (M.C.);

**Keywords:** Rathke’s cleft cysts, management model, cystic sellar lesion, headache, transsphenoidal surgery

## Abstract

**Background:** Rathke’s cleft cysts (RCCs) are non-neoplastic and rare sellar lesions derived from remnants of Rathke’s pouch. While asymptomatic RCCs often require only conservative management, symptomatic cases may necessitate surgical intervention. The aim of our study is to investigate the correlations between clinical, radiological and histological features of RCCs to propose a novel management model. **Methods**: We conducted a retrospective analysis from patients who underwent surgery for cystic lesions between 2013 and 2023 in the Neurosurgery Department of Treviso Hospital (Italy) using for our purpose only those confirmed by histological specimen as RCCs. **Results:** 20 patients for a total of 24 procedures (20 primary surgeries/4 cases for recurrence) were performed for RCCs. A gross total resection was achieved in 23 cases, resulting in improvement of headache and visual symptoms in all patients. Hyperintensity on T1-weighted MRI sequences is more strongly correlated with inflammation and with intralesional metaplasia (*p* = 0.009). Both characteristics are involved in the development of hypopituitarism (*p* = 0.057), headache, and visual impairment (*p* = 0.082) compared to cysts with CSF-like content, even when the latter are smaller in size (*p* = 0.078). Discussion and **Conclusions:** RCCs are rare lesions whose management is challenging due to a lack of established guidelines. Intraoperative cystic content and MRI cystic characteristics seem to correlate with clinical presentation and long-term outcome in these patients. The transsphenoidal endoscopic approach is a safe and effective treatment, especially in cysts with inflammatory aspect in histopathological specimens and in dedicated MRI sequences that could take advantage of an early surgical resection. A decision-making model based on clinical, radiological and histopathological features of cysts could be useful to guide RCCs’ treatment, underlining the role of inflammation that seems to be involved in the onset of visual and hormonal impairment and in recurrence risk.

## 1. Introduction

Rathke’s cleft cysts (RCCs) are benign, non-neoplastic, epithelial-lined cystic lesions arising from remnants of Rathke’s pouch, a structure involved in the embryological development of the anterior pituitary gland [1,2]. These cysts account for approximately 1% of all primary intracranial tumors and are the second most common lesions affecting the sellar and suprasellar regions [3]. The reported incidence of RCCs varies between 11.3% and 22% based on radiological and autopsy studies, suggesting that many cases remain asymptomatic and undiagnosed [4,5].

The clinical presentation of RCCs is highly variable, largely depending on their size, location, and the involvement of surrounding structures [6,7]. While these cysts are often incidental findings on neuroimaging, symptomatic cases typically present between the fourth and fifth decades of life [6]. The most common symptoms include headaches, which occur in 25–93% of cases, visual disturbances in 13–48%, hormonal dysfunction in 19–73%, and diabetes insipidus in 0–20% [1]. In rare instances, RCCs can present acutely with a clinical syndrome resembling pituitary apoplexy due to sudden cyst expansion, hemorrhage, or rupture [8].

Endocrine dysfunction associated with RCCs is primarily due to compression of the normal pituitary gland and stalk, leading to hypopituitarism [9]. It is well established that inflammatory processes within RCCs may further contribute to glandular damage and endocrine deficits. The presence of suprasellar extension and cyst enlargement increases the likelihood of visual impairment due to optic chiasm compression, making surgical intervention necessary in many cases. However, in asymptomatic patients or those presenting with isolated headaches, the decision to proceed with surgery remains controversial and is often guided by factors such as cyst size, rate of progression, and the risk of future complications [10,11].

Surgical management of RCCs is primarily performed through an endoscopic endonasal transsphenoidal approach, which allows direct access to the sellar and suprasellar regions with minimal invasiveness [12,13]. While this approach is generally associated with favorable outcomes, recurrence remains a significant concern, particularly in cases where subtotal cyst resection or simple fenestration is performed [14]. Factors such as squamous metaplasia, incomplete resection, and suprasellar extension have been implicated in RCC recurrence, underscoring the importance of long-term follow-up and individualized management strategies [15].

For these reasons, clinical presentation, surgical outcomes of RCCs, and their management remain challenges for neurosurgeons and endocrinologists alike [16]. To date, there is a lack of well-established guidelines or standardized algorithms for Rathke’s cleft cysts management. Herein, we conducted a retrospective analysis of patients who underwent surgery for RCCs at the Neurosurgical Unit of Treviso Hospital (Italy) over the past decade with the aim to contribute to the development of a structured management model to support clinical and surgical decision-making in the management of Rathke’s cleft cysts. By proposing a simple and useful framework for RCC management in daily clinical practice, we seek to reduce the gap in diagnostic and therapeutic approaches and to ameliorate diagnostic accuracy, therapeutic strategies and patient outcome.

## 2. Materials and Methods

### 2.1. Study Population

In this retrospective study, we analyzed a homogeneous study population of adult patients (age ≥ 18 yr) who underwent surgery for RCCs at the Neurosurgical Unit of Treviso Hospital (Italy) between 2013 and 2023. Clinical, surgical and radiological data were recovered by using the hospital’s surgical and clinical digital records system (Digistat^®^, v2.7.8 and Talete^®^, v4.1.0). The search was conducted using the following keywords: “sellar cystic lesion”; “Rathke cyst”; “sellar cyst”. From the retrieved results were included patients affected by Rathke’s cleft cyst, with or without suprasellar extension, who underwent surgery in the last ten years, as confirmed by histological findings. Cases with uncertain or not defined histological diagnosis were excluded. Given the retrospective nature of the study, specific ethical approval was not deemed necessary. Each patient provided informed consent, and the procedures were carried out in accordance with the standards of the local ethics committee and the Helsinki Declaration of 1975, revised in 1983. Data on demographics, tumor type, surgical procedure performed, complications, early postoperative course, and follow-up were collected

### 2.2. Pre-Operative Radiological and Clinical Assessment

All patients underwent a targeted preoperative pituitary MRI with and without gadolinium contrast (Gd). Post-contrast T1- and T2-weighted MRI sequences were used to evaluate the following parameters: cyst size, neural structure involvement, wall enhancement rim and intracystic nodules. For each case signal intensity of the cystic content were evaluated on non-contrast T1- and T2-weighted sequences with the aim to provide a correlation with the intraoperative cystic content, in accordance with MRI intensity in both the coronal and sagittal planes. Additionally, preoperative imaging included a thin-slice CT scan of the paranasal sinuses to assess the surgical route of endonasal approach. Preoperative evaluations involved ENT ophthalmological assessments and hormonal axes.

### 2.3. Surgery and Postoperative Radiological and Clinical Follow-Up

All patients underwent an endoscopic endonasal transsphenoidal approach with a unilateral paraseptal route. RCC diagnosis was confirmed by histopathological examination, and cystic content was categorized intraoperatively according to its density and composition. A postoperative brain CT scan was performed for all patients to rule out any surgical complications and to assess the extent of cyst resection. A follow-up MRI of the pituitary with gadolinium contrast was conducted at 6 months post-surgery, annually for the next five years, and biennially for the subsequent five years. Clinical follow-up included periodic endocrinological, ENT, and ophthalmological evaluations, coordinated with the timing of neuroradiological assessments.

### 2.4. Statistical Analysis

Statistical analyses were performed using R^®^ software version 3.6.3 (http://www.R-project.org) and SPSS^®^ Statistics Subscription, build 1.0.0.1327 (IBM Inc.^®^, Armonk, NY, USA).

For continuous numerical variables, measures of central tendency (mean and median) and dispersion (standard deviation) were calculated. When analyzing only two independent variables, comparisons were conducted using Student’s *t*-test, while comparisons among more than two variables were performed using one-way ANOVA. Post hoc analysis was not required. Given the limited sample size of this study, a bilateral significance level of *p* < 0.10 was applied.

For categorical (dichotomous) variables, Fisher’s exact test was employed, with statistical significance set at *p* < 0.10. Additionally, odds ratios and multivariate analyses for dichotomous variables were computed using Firth’s logistic regression, with the same *p* < 0.10 significance threshold. MRI signal characteristics were dichotomized into two categories based on T1- and T2-weighted sequences. T1 hypointensity and T2 hyperintensity were indicative of CSF-like content, whereas T1 hyperintensity and T2 hypointensity suggested a non-liquid content.

### 2.5. Statistical Analysis Considerations

The methodological adjustment to adopt a *p*-value threshold of *p* < 0.10, rather than the conventional *p* < 0.05, reflects the need to adapt statistical criteria to the constraints of the available data due to the small size of the cohort, a common challenge in studies about rare disease such as Rathke’s cleft cyst. In this context, a *p*-value threshold of *p* < 0.10 increases the likelihood of false positives, but in studies with small sample size, where statistical power is inherently limited, this lowering threshold can be justified to enhance sensitivity and reduce the risk of overlooking potential association without a loosening of statistical rigor. In this way, it is possible to underline potential useful findings that, despite not reaching conventional statistical significance, could still provide valuable preliminary insights.

## 3. Results

From a total population of 329 patients underwent surgery for sellar lesions, surgical records identified 31 cystic lesions. After histological confirmation, 20 patients with histological diagnosis of RCCs were included in this study (Figure 1). Demographic, radiological, clinical and histopathological data are summarized in Table 1. In four patients RCCs were an incidental finding.

### 3.1. Surgical Results, Recurrences and Long-Term Follow-Up

We performed twenty-four endoscopic endonasal paraseptal trans-sphenoidal approaches, twenty as a first surgery and four surgical procedures for recurrence. In one patient RCCs relapsed twice. A total resection of the cyst was achieved in twenty-three cases and in one case a partial cyst wall resection and emptying of cyst was obtained. Headaches improved in all cases with a complete resolution in eleven cases and a partial relief in three cases. Visual defects improved in all cases with a complete resolution of visual defect in three cases and a partial improvement in the remaining four cases.

The serum level of prolactinemia improved in 10 cases. In seven patients there was a transient worsening of the anterior pituitary function, while in three patients a permanent worsening occurred. In case of preoperative hormonal deficiency, no improvements were detected. Among two patients that had preoperative diabetes insipidus (DI), one resolved postoperatively. Six patients developed a transient postoperative diabetes insipidus. Indeed, in one patient a postoperative permanent DI developed. In three patients a postoperative CSF nasal leak occurred, and all these cases required reconstructive surgery. Intraoperative findings of RCCs are summarized in Table 2.

Long term follow-up was available for 19 patients with a range from 12 to 120 months with a median of 67.8 months. One patient deceased during follow-up for reasons not related to the RCCs. In our series four relapses in three patients occurred (21%) with an average time free of relapse of 9 months (range from 5 to 16 months). A second surgery was performed in two patients and in another one a recurrence occurred twice and required a second and third surgery. The average time between the first surgery and second one was 48 months (range: 9–96 months).

### 3.2. Clinical, Radiological, and Histopathological Correlations

According to our statical analysis, the correlations between clinical presentation, radiological characteristics, and histopathological features of Rathke’s cleft cysts (RCCs) revealed several significant associations.

Headache and hyperintensity on T1-weighted MRI sequences revealed significant association (*p* = 0.082), as hormonal deficiencies were significantly associated with hyperintensity on T1-weighted MRI sequences (*p* = 0.057) and with a strong correlation with the presence of inflammatory components and/or squamous metaplasia (*p* = 0.009). Notably, panhypopituitarism was more prevalent in patients with RCCs exhibiting hyperintense signals on T1-weighted MRI sequences (*p* = 0.025) and was significantly associated with an intrasuprasellar localization (*p* = 0.043). Additionally, suprasellar extension demonstrated a stronger correlation with panhypopituitarism than with isolated hormonal deficits (*p* = 0.091).

Diabetes insipidus was significantly associated with the presence of squamous metaplasia (*p* = 0.071). Incidentally detected RCCs exhibited a statistically significant correlation with intrasellar localization (*p* = 0.008), smaller cyst dimensions (<15 mm) (*p* = 0.077), and isointensity on T1-weighted MRI sequences (*p* = 0.010).

An evaluation of histopathological radiological correlations demonstrated that cystic content with a cerebrospinal fluid (CSF)-like appearance was significantly associated with hypointensity on T1-weighted MRI sequences (*p* = 0.024) and hyperintensity on T2-weighted MRI sequences (*p* = 0.007), in alignment with findings reported in the literature [3,4]. In contrast, mucoid cystic content was correlated with isointensity on T2-weighted MRI sequences (*p* = 0.018) and, to a lesser extent, with hyperintensity on T2-weighted MRI sequences (*p* = 0.036). Furthermore, mucoid cysts more frequently exhibited iso/hypointense signals on both T1-weighted (*p* = 0.071) and T2-weighted MRI sequences (*p* = 0.022) compared to cysts with CSF-like content.

Cysts with purely granular or mixed (mucoid/granular) content were significantly correlated with hyperintensity on T1-weighted MRI sequences (*p* = 0.074) and with the presence of inflammation and/or squamous metaplasia (*p* = 0.056). Inflammatory changes, with or without squamous metaplasia, were strongly associated with hyperintensity on T1-weighted MRI sequences (*p* = 0.007). Additionally, inflammatory processes correlated with an inhomogeneous signal on T2-weighted MRI sequences (*p* = 0.042), whereas squamous metaplasia was associated with hypointensity on T2-weighted MRI sequences (*p* = 0.088) and hyperintensity on T1-weighted MRI sequences (*p* = 0.005).

Among patients presenting with headaches, cysts with CSF-like content were, on average, larger than those with mucoid or granular content (*p* = 0.078).

Furthermore, an analysis of correlations between cyst size, anatomical extension, histopathological features, and intraoperative/postoperative cerebrospinal fluid (CSF) leakage identified significant associations. Lesions exceeding 20 mm in diameter were significantly associated with an increased risk of postoperative CSF leakage (*p* = 0.040). Moreover, intraoperative CSF leaks attributed to arachnoid fissures were significantly correlated with postoperative CSF leakage (*p* = 0.047). The development of new postoperative hormonal deficiencies was significantly associated with hyperintensity on T1-weighted MRI sequences (*p* = 0.043).

No statistically significant correlations were identified between patient sex, clinical presentation, radiological features, intraoperative histological findings, and postoperative outcomes. Likewise, no significant associations were observed between RCC anatomical characteristics and intraoperative CSF leakage. Finally, no statistically significant correlation was found between postoperative CSF leakage and cystic content (Table 3 and Table 4).

These findings allowed us to propose a decision-making model based on clinical and radiological criteria in guiding the surgical management of RCCs as follows:RCCs > 10 mm with hyperintensity on T1-weighted MRI sequences should be considered for surgical intervention, given their higher likelihood of progressing to pituitary insufficiency.RCCs > 10 mm with hypointensity on T1-weighted MRI sequences should undergo annual radiological and visual field follow-up, with surgery considered in cases of cyst enlargement or emerging visual deficits.RCCs ≤ 10 mm with hyperintensity on T1-weighted MRI sequences should be monitored radiologically, with surgery reserved for cases of cyst growth.RCCs > 10 mm presenting with isolated hyperprolactinemia and hyperintensity on T1-weighted MRI sequences may warrant surgical intervention, given the potential for progression to hypopituitarism or visual impairment (Figure 2).

## 4. Discussion

Although RCCs are often asymptomatic or incidentally detected, they may present with severe symptoms, such as headaches or visual impairment, due to their variability in natural history and their histopathological and radiological heterogeneity [17].

Our proposed model computed the several correlations found between radiological and clinical features of Rathke’s cleft cysts in our series to improve their treatment with a patient-tailored strategy. A first important aspect that emerged from our study is the strong association between RCCs and endocrine dysfunction, underlining the impact of hormonal evaluation in these patients. Specifically, hypopituitarism and headaches were observed in patients with cysts hyperintense on T1-weighted MRI sequences subsequently correlated to chronic inflammation, squamous metaplasia at histopathological investigation and correlated to intraoperative granular or mucoid elements cystic content. These findings align with previous studies [18] confirming the paramount role of inflammation and squamous metaplasia in the development of postoperative hypopituitarism. Notably, hypopituitarism did not improve with surgery, and new postoperative hormonal deficits were observed in 16% of patients in our series. Conversely, RCCs with a CSF-like content (hyperintensity on T2-weighted MRI sequences and hypointensity on T1-weighted MRI sequences) demonstrated more indolent cystic behavior and a lower incidence of inflammation on histopathological specimens [7] (Figure 3 and Figure 4).

Another key point that emerged from the analysis is a strong concordance between the preoperative MRI characteristics of RCCs and their intraoperative findings. In detail, CSF-like cystic content is significantly associated with hypointensity on T1-weighted MRI sequences (*p* = 0.024) and hyperintensity on T2-weighted MRI sequences (*p* = 0.007); in contrast, mucoid cysts were more frequently isointense on T2-weighted MRI sequences (*p* = 0.018). Consequently, the intensity on MRI sequences accurately reflected the intraoperative cystic content allowing them to provide preoperative insight about the histopathological aspect of RCCs which may influence both the surgical plan and postoperative prognosis.

Several studies have highlighted the possible value of MRI and molecular or blood-based biomarkers in assessing the behavior of cysts [3,19]. However, most of these studies are based on small population cohorts, which often leads to unclear findings. Moreover, while the investigation of blood-based inflammatory indices and molecular biomarkers was not within the aim of the present study, it is noteworthy that previous research has generally failed to establish their effectiveness in providing reliable indicators of inflammation or recurrence risk [18,20]. Moreover, other Authors [21] described the use of MRI spectroscopy as an alternative exam to investigate the behavior of cysts in term of aggressiveness and tendency to relapse but, even if interesting, its application remains complex in daily clinical practice and confined to experimental protocols due to the complex anatomy of the sellar region that includes in a small space several neural and vascular structures. These considerations underscore the complexity of establishing robust biomarkers for clinical use in RCCs.

Conversely, an interesting finding that emerged from our study is that postoperative CSF leakage and cyst size were significantly associated with cysts larger than 20 mm in diameter. This finding is consistent with previous reports [22], suggesting that cyst size plays a pivotal role in surgical risk stratification.

However, even if our study did not identify a precise cut-off for cyst size, according to our experience, a threshold of 10 mm of diameter appears to be a critical point in the surgical management of RCCs. In fact, cysts greater than 10 mm increase the likelihood of a mass effect and compression of adjacent neural structures, such as the pituitary gland and optic chiasm. Indeed, cysts larger than 12 mm are frequently associated with headaches, visual impairment and endocrine dysfunctions. In our series only cysts larger than 10 mm were taken into consideration for surgery. Unlike RCCs smaller than 10 mm, these are often asymptomatic and an incidental finding. Also, the risk of recurrence and progression is influenced by cyst size with cysts exceeding 10 mm exhibiting a higher tendency to recur if not completely resected. Additionally, squamous metaplasia has been more frequently observed in RCCs larger than 10 mm and linked to increased recurrence risk [18].

For these reasons, the decision and timing of surgery should be based on the anatomical characteristics of cysts as size, optic pathway compression and the presence of severe symptoms, such as visual impairment or refractory headaches.

While, in the recent literature, the different viable options of treatment are an object of debate for asymptomatic or minimally symptomatic RCCs [17], surgical intervention remains the treatment of choice for visual symptoms [23,24] and, in this scenario, our model may help to stratify the patients affected by RCCs for a conservative versus a surgical management. In our series, all patients undergoing surgery for visual impairment experienced postoperative visual improvement without complications. In addition to visual deterioration, headaches and RCC characteristics also play an important role in the decision-making of these patients. Our series showed an interesting observation that CSF-like cysts with hyperintensity on T2-weighted MRI sequences caused headaches only when they reached a larger size (>20 mm). The lack of inflammation in these cysts may explain this pattern. In contrast, cysts smaller than those with CSF-like content but with hyperintensity on T1-weighted MRI sequences were more frequently associated with headaches and hypopituitarism, suggesting a different pathophysiological mechanism. These findings may also assist in determining surgical candidacy in RCCs detected incidentally, utilizing both size criteria and cyst content assessment on MRI.

However, a correlation between cyst size, visual impairment, and chiasmal contact did not reach statistical significance, suggesting that other factors, such as inflammation, may contribute to visual impairment. Additionally, cyst growth rate may play a role in visual deterioration, especially when cyst growth exceeds 3 mm per year as some Authors described [25].

A surgical approach in the management of RCCs is supported by advancements in endoscopic transsphenoidal surgery over the past three decades [13,26] that have significantly improved its risk profile in surgery as well as in the initial phase of disease. The early removal of small cysts, particularly those around 10 mm, is associated with a low risk of cerebrospinal fluid leakage, a complication strongly correlated with cyst size. Indeed, an early surgical approach may prevent complications related to mass effects, especially in cysts with a high potential for progression, in addition to being technically less complex, granting a complete resection of cystic wall with respect to RCCs with suprasellar extension. Moreover, longstanding RCCs are more likely to exhibit squamous metaplasia that is associated with increased recurrence risk [18]. A safety profile of endoscopic endonasal approach is also shown by small rate complications in our series.

RCCs have a tendency to recur, especially those with squamous metaplasia and incomplete wall resection, as underlined in previous studies [27]. Our series identified the first signs of relapse within 5–16 months postoperatively (mean: 9 months) in 21% of cases with a mean long-term follow-up of 67.8 months. However, in line with a recent meta-analysis [28] that described a second surgical procedure for relapse beyond 72 months, in our research a reintervention for relapse became necessary approximately at a mean of 48 months after first surgery. Given these findings, the Kaplan–Meier curve extracted from our series confirms that a long-term follow-up (>5 years) is essential for RCC patients (Figure 5). This survival analysis provides a valuable insight into the natural history of Rathke’s cleft cyst post-surgery, highlighting the importance of early surveillance in the postoperative period to identify first signs of recurrences to guide patient counseling and clinical follow-up strategies.

Although our investigation did not establish a significant correlation between squamous metaplasia and recurrence, likely due to the small sample size, the development of early signs of relapse may be predicted by intraoperative signs of inflammation and by squamous metaplasia in histological specimen of cysts which have relapsed, as underlined by other Authors [27,28]. In cases of recurrence, surgical indications should be based on the same criteria applied for the initial procedure, especially the size of the cyst, mass effect and visual impairment.

All these considerations, along with the improved safety and efficacy of endoscopic transsphenoidal techniques [17], encourage early surgical intervention in select cases, offering a low surgical risk and a minimal potential for long-term complications. However, due to the lack of established guidelines and the variable biological behavior of cysts, different aspect in MRI and the variable histopathological patterns of RCCs, the patient selection should remain highly tailored, balancing the surgical risks against the potential cyst growth and consequent greater morbidity. To the best of our knowledge, even if various research groups have proposed radiological classification systems and decision-making algorithms [26] to predict the biological behavior of RCCs useful for a surgical strategy, there is no universally accepted agreement for the management of RCCs. The most established surgical indications include visual deficits, progressive hypopituitarism, and severe symptoms such as refractory headaches.

In our opinion, a tailored approach integrating radiological findings, clinical presentation, cyst size, and histopathological patterns is the most useful strategy. In fact, our study has as key points the correlation between T1 signal intensity, the intraoperative presence of inflammation, histopathological presence of squamous metaplasia and the clinical symptoms to differentiate RCCs with a more aggressive pattern from indolent forms. Hyperintensity on T1-weighted imaging associated with inflammatory aspects or squamous components may predict a more unfavorable course, suggesting a proactive surgery even in cases of cysts without mass effect symptoms. Among the most recent classification systems, Kim et al. [19] categorized RCCs into different subgroups (Type S-low, S-iso, S-high and Type M) based on T1 and T2 signal intensity, demonstrating variable propensities for growth and symptom development. Although, our study does not adopt a rigid group based on T1 and T2 MRI signal patterns, it highlights the clinical significance of hyperintensity in T1-weighted sequences that often are associated with a denser or inflammatory cystic content and are correlated with headaches and pituitary dysfunction. This aligns with the underlying principle that radiological features may help to predict aggressiveness and symptomatic progression, thus guiding surgical decisions.

Another recent study by Alsavaf et al. [25] focused on RCC growth patterns, proposing a risk score system based on MRI intensity and clinical features (age, smoking status) to stratify patients at higher risk of progression and, consequently, in need of early surgery. Conversely, our proposed model does not employ a numerical risk score, but observed that inflammatory content, often associated with T1-weighted sequences, may predict both the onset of pituitary deficits and a higher likelihood of recurrence supporting the value of MRI aspects as a significant factor to propose an earlier surgical intervention. Despite methodological differences, both approaches share the common objective of identifying patients at greater risk of symptomatic progression and advocating for timely intervention.

Finally, the strength of our model is based on simple criteria to investigate in a clinical practice, such as the size of the cyst, T1 signal intensity and symptoms, such as hormonal and visual defects.

## 5. Limitations of the Study

Despite the clinically relevant findings, our study presents several limitations. The most important is the small sample size that limits the statistical power of our findings, particularly in the subgroup analyses. Larger, multi-center studies are necessary to validate our observations and model to establish more precise management criteria. Another weak point of our study is its retrospective design. The retrospective nature of the study introduces potential selection and information bias. A prospective study design with standardized follow-up intervals would provide more robust conclusions.

Therefore, from a clinical point of view our data is lacking in long-term endocrine follow-up. In fact, although we identified a correlation between RCC inflammation and hypopituitarism, long-term endocrine outcomes remain unclear. Future studies with extended follow-up are needed to assess whether surgical intervention alters the progression of pituitary dysfunction over time.

## 6. Conclusions

RCCs are a rare and challenging condition with a complex management approach. Our findings align with existing studies in demonstrating that specific radiological criteria, particularly T1 hyperintensity, and clinical symptoms, primarily visual impairment, are angle stones for surgical decision-making, whereas asymptomatic or mildly symptomatic cases may be safely monitored. With respect to other Authors [19,21,26], our model is more focused on the role of inflammation and its correlation with onset symptoms, cyst size and MRI criteria.

If the model is validated by an additional larger series, its simplicity makes it a highly practical tool in daily clinical practice.

## Figures and Tables

**Figure 1 diagnostics-15-00886-f001:**
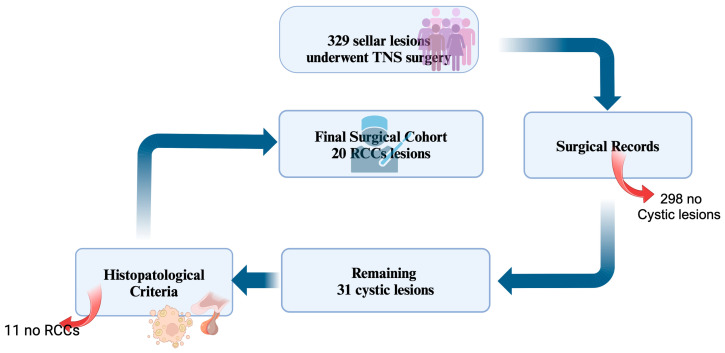
Population study selection. Created by Biorender^®^ (http://app.biorender.com, 23 February 2025).

**Figure 2 diagnostics-15-00886-f002:**
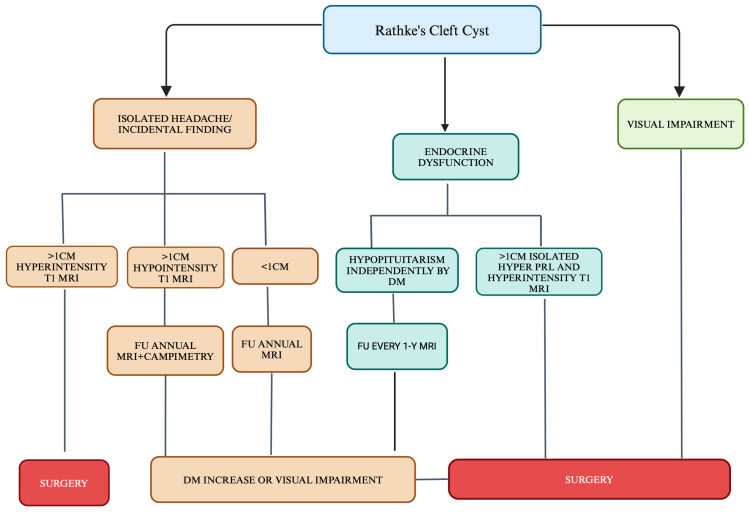
Proposed model for RCC management. Created by Biorender^®^.

**Figure 3 diagnostics-15-00886-f003:**
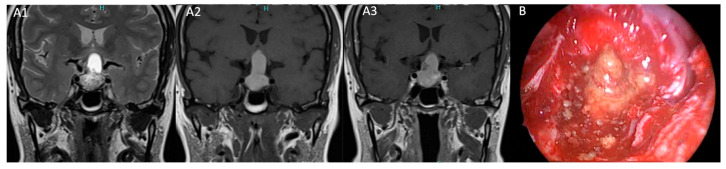
(**A**,**B**): Pituitary MRI and intraoperative view of patient 16. (**A1**) Preoperative T2-weighted of targeted pituitary MRI in coronal plane show a sellar–suprasellar cyst characterized by hyperintensity of signal in the cranial content and inhomogeneous signal in the sellar portion. (**A2**) Hyperintensity of signal on T1-weighted MRI sequence. (**A3**) No contrast enhancement on T1-weighted with Gadolinium of pituitary MRI sequence. (**B**) The intraoperative view shows the granular content of the cyst. This patient experienced headache, visual defects and hypopituitarism.

**Figure 4 diagnostics-15-00886-f004:**
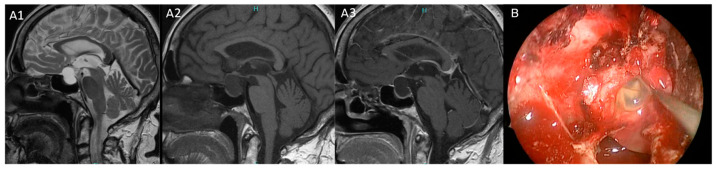
(**A**,**B**): Pituitary MRI and intraoperative view of patient 18. (**A1**) Preoperative T2-weighted of targeted pituitary MRI in sagittal plane that show a sellar–suprasellar cyst characterized by hyperintense of signal; (**A2**) Hypo intensity of signal on T1-weighted without gadolinium MRI sequence. (**A3**) T1-weighted sequence with Gadolinium of targeted pituitary MRI. (**B**) The intraoperative view shows the mucoid content of cyst. This patient presented with amenorrhea and visual field defect.

**Figure 5 diagnostics-15-00886-f005:**
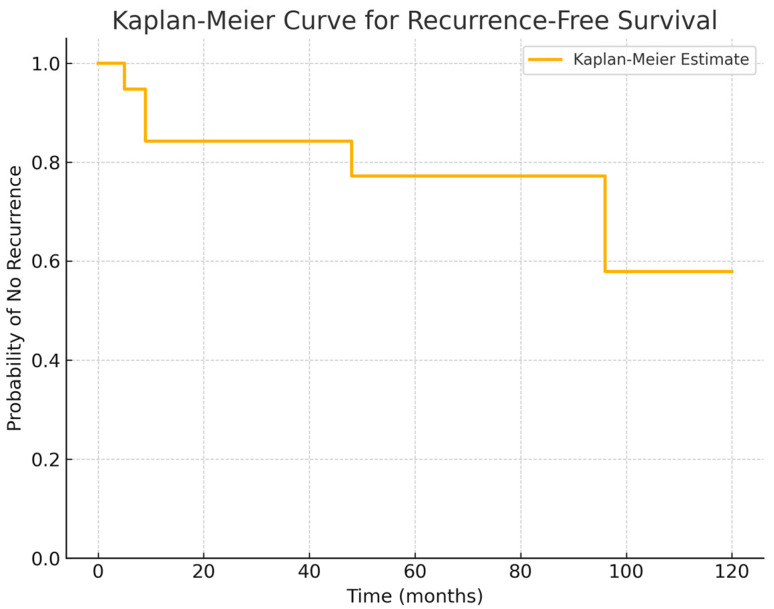
Kaplan–Meier curve suggests that the median time to recurrence is within the first-year post op, with recurrences occurring between 5 and 16 months in the subset of patients who developed recurrence. Notably, beyond the first 24 months, the probability of recurrence-free survival stabilizes, suggesting that most recurrences occur relatively early in the follow-up period. However, long-term follow-up remains essential, given the potential for late recurrences.

**Table 1 diagnostics-15-00886-t001:** Demographical, clinical radiological and histopathological population data.

Patients	Age, Sex (Years)	Clinical Presentation	Hormonal Axes Involvment	DM (mm)	MRI T1 Intensity	MRI T2 Intensity	Cystic Content	Histology
1	51, F	H	-	27	Hypo-	Hyper-	L	Ab
2	70, F	H+V+Hy	Hp+C+T+GH	21	Hypo-	Hyper-	L	Ab
3	65, M	H+Hy	Hp+C+T+Hg	19	Hyper-	Hypo-	G	Sm
4	59, F	I	-	21	Iso-	Hyper-	L	Ab
5a	38, F	H+V	Hp	15	N/A	N/A	G	Ab
5b (I relapse)	46	H+Hy+DI	C+T+Hg	12	Hyper-	Iso-	M	Sm
6	57, F	H	-	21	Hypo-	Hyper-	n/a	Ab
7a	79, F	V+Hy+P	Hp+C+Hg+GH	21	Hypo-	Hyper-	n/a	Ab
7b (I relapse)	80	V+Hy+P	Hp+C+Hg+GH	18	Hypo-	Hyper-	L	Ab
8	55, F	I	T	15	Hypo-	Hyper-	L	In
9	39, F	H	-	21	Iso-	Iso-	M	Ab
10	54, M	I	-	11	Iso-	Hyper-	L	Ab
11	73, F	V+Hy	C+Hg	23	Hyper-	Inh-	M	In+Sm
12	73, M	H+Hy	Hp+C+T	17	Hyper-	Inh-	Mx	In
13	45, F	H+A	Hp	25	Hyper-	Hyper-	L	Ab
14	70, F	Hy	Hp+C+T+Hg+GH	19	Hyper-	Hypo-	G	Sm
15	64, F	I	Hp	14	Iso-	Iso-	L	Ab
16	44, F	A+V	Hp	21	Hypo-	Hyper-	M	In
17	68, F	Hy	C+T+Hg+GH	32	Iso-	Hyper-	G	Ab
18	58, F	H+V+Hy	Hp+C+Hg	27	Hyper-	Inh-	G	In
19a	36, F	H+V+Hy+A	Hp+C+T+GH	15	Hyper-	Hyper-	n/a	Ab
19b (I relapse)	40	H+Hy+A	Hp+C+T+Hg+GH	N/A	Hypo-	Hyper-	G	In
19c (II relapse)	43	H+Hy+A	C+T+Hg+GH	N/A	Hyper-	Hyper-	G	In+Sm
20	29, F	H+V+Hy+A+DI	Hp+C+Hg+GH	15	Iso-	Hyper-	G	In+Sm

Abbreviations: (H) Headache; (A) oligo/amenorrhea; (V) visual defects; (Hy) Hypopituitarism; (DI) diabetes insipidus; (I) incidental findings; (Hp) hyperprolactinemia; (C) hypercortisolism; (T) hypothyroidism; (Hg) hypogonadism; (GH) GH deficiency; (N/A) not available data; (-) normal hormonal axes; (Hypo-) Hypointensity; (Hyper-) Hyperintensity; (Iso-) isointensity; (Inh-) Inhomogeneous; (L) liquid; (M) mucoid; (G) granular; (Mx) mixed; (In) inflammation; (Sm) squamous metaplasia; (Ab) absence of inflammation and/or squamous metaplasia.

**Table 2 diagnostics-15-00886-t002:** Intraoperative and histopathological findings.

Intraoperative and Histopathological Findings	*N* (%)
Intra op CSF leak	7 (29%)
Arachnoidal fissure	5
Recognizable pituitary gland	11 (46%)
Cystic content	
- CSF like	8 (38%)
- Mucoid/gelatinous	4 (19%)
- Granular	8 (38%)
- Mixed	1 (5%)

**Table 3 diagnostics-15-00886-t003:** Statistical correlation between intensity of signal on T1- and T2-weighted MRI sequences, symptoms, cystic content and histopathological findings.

	Headache	Hypopituitarism	Granular Content	CSF Like Content	Inflammation/Squamous Metaplasia
MRI Hyperintensity on T1 weighted	*p* = 0.082	*p* = 0.025	*p* = 0.074	-	*p* = 0.007
MRI Hyperintensity on T2 weighted	-	-	-	*p* = 0.007	-
Inflammation and/or squamous metaplasia	-	*p* = 0.009	*p* = 0.056		

Note: (-) non statistical correlation. *p* value ≤ 0.10.

**Table 4 diagnostics-15-00886-t004:** Statistical correlation between the dimension, content and clinical presentation of Rathke’s cleft cyst series with or without headache.

Symptoms	CSF Like Content	Mucoid/Granular Content	*p*-Value
	Mean Diameter (mm) ± σ		
Headache	24.3 ± 3.0	18.0 ± 4.9	*p* = 0.078
Not headache	17.2 ± 4.3	22.0 ± 7.6	*p* = 0.267
	*p* = 0.046	*p* = 0.312	

Note: *p* value ≤ 0.10.

## Data Availability

Manuscript data are embedded in the text and fully available on specific request.

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
