# Peer review of "A Data-Driven Approach for Rathke’s Cleft Cysts Management"

_diagnostics, 2025, doi:10.3390/diagnostics15070886_

Round 1

Reviewer 1 Report

Comments and Suggestions for Authors

While the study provides an interesting correlation between MRI features and Rathke’s cleft cyst management, the conclusions regarding surgical decision-making and MRI-based predictions require further validation in larger, prospective studies.

  1. There are grammatical errors and awkward phrasing that reduce readability (e.g., “Infact was observed a strong association…” should be “In fact, we observed a strong association…”).
  2. Some sections are redundant and repetitive, particularly in explaining MRI characteristics and their clinical relevance.
  3. The proposed management algorithm lacks clear decision-making thresholds for when to operate versus monitor patients
  4. The study states that RCCs >10 mm with hyperintensity on T1-weighted MRI should be considered for surgery, but this is based on weak statistical evidence (p = 0.057).
  5. The risk-benefit ratio of early surgical intervention is not fully discussed, especially for asymptomatic or minimally symptomatic patients.
  6. The average recurrence-free period is only 9 months, while prior studies suggest RCC recurrence often occurs beyond 5 years.
  7. It is unclear whether the observed MRI characteristics are reliable biomarkers for inflammation or recurrence risk, as the study does not compare with alternative diagnostic tools like spectroscopy or biomarker analysis.
  8. Many statistical comparisons use a p-value threshold of 0.10, which is weaker than the conventional p < 0.05 standard, increasing the risk of false-positive findings.
Comments on the Quality of English Language

There are grammatical errors and awkward phrasing that reduce readability (e.g., “Infact was observed a strong association…” should be “In fact, we observed a strong association…”).

Reviewer 2 Report

Comments and Suggestions for Authors

The authors present a retrospective analysis of Rathke’s cleft cysts (RCCs) and propose a novel management model based on clinical, radiological, and histopathological features. The study is well-structured and provides comprehensive data offering valuable insights into RCCs. The introduction is informative, the methodology is clearly defined, and the statistical analysis is generally appropriate. However, there are a few minor concerns that should be addressed to enhance clarity and strengthen the discussion.

Specific Comments

  • The abstract provides a good summary but could be more concise. The conclusions should explicitly state the clinical significance of the findings in a clearer manner.
  • The introduction includes a strong background on RCCs but lacks a clear research question. The study's objective should be more explicitly emphasized.
  • Statistical Significance (p-value): The study uses a p < 0.10 threshold for statistical significance, which is higher than the conventional p < 0.05. A brief justification for this choice would be beneficial.
  • While the results discuss recurrence rates, a Kaplan-Meier survival analysis for recurrence-free survival would be useful for better visualization.
  • The discussion adequately relates findings to the existing literature, but causal interpretations should be avoidedin a retrospective study. For instance, the statement:

"Inflammation plays a crucial role in the development of hypopituitarism."
Should be softened to:
"Inflammation is strongly associated with hypopituitarism, suggesting a potential role in its development."

  • References: The manuscript should incorporate larger series and more recent studies to enhance its credibility. The following references should be included:
  1. Emengen A, Gokbel A, Uzuner A, et al. Surgical Strategies Regarding the Extended Endoscopic Transnasal Approach for Isolated Suprasellar Rathke's Cleft Cysts. World Neurosurg. Published online February 28, 2025. doi:10.1016/j.wneu.2025.123757
  2. Hacioglu A, Tekiner H, Altinoz MA, et al. Rathke's cleft cyst: From history to molecular genetics. Rev Endocr Metab Disord. Published online February 13, 2025. doi:10.1007/s11154-025-09949-6
  3. Matsushita S, Shimono T, Maeda H, et al. Comparison of clinical and radiological characteristics of inflammatory and non-inflammatory Rathke cleft cysts. Jpn J Radiol. 2025;43(1):32-42. doi:10.1007/s11604-024-01641-0
  4. Punukollu A, Franklin BA, Pineda FG, et al. Efficacy and safety of surgical management for Rathke's cleft cysts in pediatric patients: a systematic review and meta-analysis. Neurosurg Rev. 2024;48(1):13. Published 2024 Dec 30. doi:10.1007/s10143-024-03156-8
  5. Cabuk B, Selek A, Emengen A, Anik I, Canturk Z, Ceylan S. Clinicopathologic Characteristics and Endoscopic Surgical Outcomes of Symptomatic Rathke's Cleft Cysts. World Neurosurg. 2019;132:e208-e216. doi:10.1016/j.wneu.2019.08.196
  • The proposed management model is promising, but how does it compare to existing guidelines or classification systems? A brief discussion on this would strengthen its clinical impact.

Final Recommendation

The manuscript is scientifically sound and provides a valuable contribution to the literature. However, the discussion requires slight refinement, some methodological clarifications should be addressed, and more extensive series and recent references should be incorporated. The authors are encouraged to revise the manuscript accordingly.

Round 2

Reviewer 1 Report

Comments and Suggestions for Authors

The article can be accepted in its current form.

Reviewer 2 Report

Comments and Suggestions for Authors

The authors have addressed the reviewers' comments and made the necessary revisions. I believe the manuscript is acceptable in its current form.